# Fabrication and Characterization of Novel Poly(d-Lactic Acid) Nanocomposite Membrane for Water Filtration Purpose

**DOI:** 10.3390/nano11020255

**Published:** 2021-01-20

**Authors:** Lau Kia Kian, Mohammad Jawaid, Salman Alamery, Ashok Vaseashta

**Affiliations:** 1Institute of Tropical Forestry and Forest Products (INTROP), Universiti Putra Malaysia, Serdang 43400, Selangor, Malaysia; laukiakian@gmail.com; 2Department of Biochemistry, College of Science, King Saud University, PO Box 22452, Riyadh 11451, Saudi Arabia; salamery@ksu.edu.sa; 3Office of Applied Research, International Clean Water Institute, Manassas, VA 20112, USA; prof.vaseashta@ieee.org; 4Biomedical Engineering and Nanotechnologies Institute, Riga Technical University, 1 Kalku Street, 1658 Riga, Latvia

**Keywords:** membrane, nanocellulose, morphology, thermal, tensile, permeability

## Abstract

The development of membrane technology from biopolymer for water filtration has received a great deal of attention from researchers and scientists, owing to the growing awareness of environmental protection. The present investigation is aimed at producing poly(D-lactic acid) (PDLA) membranes, incorporated with nanocrystalline cellulose (NCC) and cellulose nanowhisker (CNW) at different loadings of 1 wt.% (PDNC-I, PDNW-I) and 2 wt.% (PDNC-II PDNW-II). From morphological characterization, it was evident that the nanocellulose particles induced pore formation within structure of the membrane. Furthermore, the greater surface reactivity of CNW particles facilitates in enhancing the surface wettability of membranes due to increased hydrophilicity. In addition, both thermal and mechanical properties for all nanocellulose filled membranes under investigation demonstrated significant improvement, particularly for PDNW-I-based membranes, which showed improvement in both aspects. The membrane of PDNW-I presented water permeability of 41.92 L/m^2^h, when applied under a pressure range of 0.1–0.5 MPa. The investigation clearly demonstrates that CNWs-filled PDLA membranes fabricated for this investigation have a very high potential to be utilized for water filtration purpose in the future.

## 1. Introduction

Membrane technology has very commonly been applied in purification, dialysis, filtration, and for pharmaceutical applications for several decades. The porous nature of membranes offers them the capability of the efficient separation of molecules or compounds, while maintaining a steady movement of water molecules across the semi-permeable membrane [1]. Polymeric materials are widely utilized for the fabrication of membranes due to their ease of processability, and also due to a recent increase in research interest by many researchers to explore optimal processing parameters and methods to produce a membrane with superior performance [2]. The most commonly used methods to fabricate polymeric membranes are by phase inversion [3,4], particulate leaching/solvent merging [5,6], freeze-drying [7] and electro-spinning [8] techniques. Phase inversion is one of the most commonly used methods and well-known in membrane fabrication. It is performed by removing the solvent from a liquid-polymer solution, leaving a porous, solid polymeric membrane. Among several potential polymeric materials, poly(lactic acid) (PLA) is a favourable candidate material for membrane fabrication, due to its stable crystallization for porosity formation, great mechanical strength, excellent thermal resistance and renewable and biodegradable nature [2,9]. Tanaka et al. [4] fabricated porous PLA membranes for microfiltration purposes by using a water-induced phase separation method. Later on, Minbu et al. [3] prepared PLA microfiltration membranes via water-induced phase separation by using a surfactant, and more recently Chinyerenwa et al. [10] produced porous PLA membranes, with phase inversion induced by hot water droplets. Recently, Gao et. al. [11] reported on the study of pore formation of PLA membranes with controlled addition of polyethylene oxide. Another investigation of formation of poly(DL-lactic acid) (PDLA) porous membranes using various co-solvents was conducted by Phaechamud and Chitrattha [12].

Recently, nano-sized additives, such as nanocellulose [13,14], carbon nanotubes (CNTs) [15,16] and zeolites [17,18] have been introduced into polymeric membrane structure for the purpose of improving their porosity, mechanical strength, thermal properties, water flux permeability, separation and anti-fouling capability. Nanocellulose, a derivative of cellulose, is an ideal nano-filler for polymeric reinforcement, due to its high stiffness and excellent heat tolerance characteristics [1]. Zhong et al. [14] recently reported on the preparation and characterization of polysulfone/sulfonated polysulfone/cellulose ternary blend membranes by enhancing its hydrophilicity properties by using cellulose nanofibers. Furthermore, Bai et al. [13] prepared a polysulfone composite membrane in higher porosity with good connectivity by the incorporation of nanocrystalline cellulose. According to the literature, varying aspect ratios of nanocellulose could form varying percolated networks, which eventually will have impact on its interaction with polymeric matrix and the void formation within such membranes. The compatibility of nanocellulose with a polymer matrix depends mainly on the Van der Waals interaction between them through hydrogen bonding [19].

In this investigation, PDLA polymer was utilized for membrane fabrication with the aid of nanocellulose acting as pore-forming agents, in addition to inducing the phase inversion process. Two types of nanocellulose, e.g., nanocrystalline cellulose (NCC) and cellulose nanowhisker (CNW), with varying aspect ratios, were employed to study their influences on the membrane porous structure formation. We note here that there is no investigation in the literature that has been reported on applying the nanocellulose biomaterial to improve the performance of the PDLA polymeric membrane. To study the effect of nanocellulose on the membrane properties, the morphology, surface wettability, thermal and mechanical analyses were carried out in this work. Additionally, the water permeability was also examined for the membranes fabricated during the investigation, in order to determine their reliability for filtration purpose.

## 2. Materials and Methods

### 2.1. Materials and Chemicals

PDLA pellets (75,000 average molecular weight, 1.6 polydispersity, 1.24 g/cm^3^ specific gravity) were procured from PTT Public Ltd., Bangkok, Central Thailand, Thailand, and were used as-purchased from supplier without any further processing to maintain uniform quality conditions. N,N-dimethylformamide (DMF), chloroform (CHCl_3_) (≥99.8% purity), sulfuric acid (H_2_SO_4_) (95–98% purity) and dialysis tube (10 mm average flat width, 2000 MWCO) were purchased from Evergreen Ltd. (Petaling Jaya, Selangor, Malaysia) in affiliation with R&M Chemicals Inc. (Subang, Selangor, Malaysia). For this study, we used DMF due to its strong dissolution characteristics for preparing highly crystalline PDLA polymer, and hence suitably used it for making good membrane structures. The NCC was produced from roselle microcrystalline cellulose (MCC), as described elsewhere in our previous paper [20]. For comparison purposes, another nanocellulose type, CNW, with a different aspect ratio, was produced according to the process treatment conditions, as described below.

### 2.2. Preparation of CNW

As a first step, we conducted hydrolysis at temperature of 45 °C to 2.5 g MCC with 80 mL of (50% wt./wt.) sulfuric acid solution, under vigorous mechanical stirring. Then, centrifugation at 6000 rpm was carried out until a pH value close to 3 was reached. After that, the resulting white cloudy suspension was subjected to ultrasonication for 15 min with an amplitude of 40% at constant 20 kHz frequency and 500 W power output. The resultant suspension was treated with dialysis in distilled water to reach a value of pH 5. Lastly, the supernatant CNW suspension was collected and freeze dried.

### 2.3. Preparation of PDLA Membranes

A 10 wt.% of PDLA pellet was mixed until fully dissolved in DMF solvent at 60 °C, under strong agitation. The polymer solution was then casted on a petri dish and cooled at room temperature for 15 min. Afterwards, it was immersed in a water coagulation bath for 5 min to induce phase inversion for the solidification process. Subsequently, the coagulated solution was subjected to freezing temperatures in a refrigerator, and then gently washed with chloroform to remove the DMF solvent. At the end, a pure PDLA membrane was obtained after drying in vacuum oven to remove remaining water and DMF contents. For nanocellulose-reinforced membranes, different amounts of NCCs and CNWs were loaded during the first mixing step, with DMF solvent at 60 °C under vigorous stirring. A series of subsequent treatment processes were then carried out, which were similar to the fabrication for pure PDLA membrane. The fabricated membranes of pure PDLA, 1 wt.% NCC filled PDLA, 2 wt.% NCC filled PDLA, 1 wt.% CNW filled PDLA and 2 wt.% CNW filled PDLA were denoted as PD-neat, PDNC-I, PDNC-II, PDNW-I and PDNW-II, respectively.

### 2.4. Nanocellulose Characterization

The isolated NCC and CNW particles were examined to have aspect ratios of 6.2 and 6.5, respectively, after analysis using Image J (1.52 Ver., National Inst. Health, Bethesda, MD, USA) software. Furthermore, the determined specific surface area for NCCs and CNWs were 18.74 m^2^/g and 19.49 m^2^/g respectively, through Brunauer–Emmett–Teller (BET) analysis using a 3Flex Surface Characterization analyzer (Micromeritics Instrument Corp., Norcross, GA, USA).

### 2.5. PDLA Membranes Characterization

A BRUKER Dimension Edge atomic force microscope (AFM) (Bruker Corp., Billerica, MA, USA) was applied to observe the structural morphology of membranes. A scan rate of 4.80–6.00 µm/s with a tapping mode was employed on the membrane samples at ambient conditions. Furthermore, the pore size of membranes was determined through AFM/Nanoscope (1.7 Ver., Bruker Corp., Billerica, MA, USA) software analysis. Apart from that, a KRUSS DSA-100 goniometer (Kruss Co., Hamburg, Germany) was used to study the wettability of membrane surface by examining the water contact angle after dropping a 2.5 µL water for 0.5 s, and this process was conducted in triplicates for each membrane sample. The heat resistance thermal stability of the samples was determined through a Q500 thermogravimetric analyzer (TGA) (TA Instruments Inc., New Castle, DE, USA) from 25–1000 °C at a 20 °C/min heating rate. An Instron 4400 Universal Tester (Instron Corp., Norwood, MA, USA) was also performed according to the ASTM D882-12 under a fixed 12.5 mm/min crosshead rate to measure the mechanical tensile for each sample. To evaluate the filtration performance, a 1000 mL volume of water was allowed to continuously flow through a cell filtration (membrane with an effective area of 16 cm^2^) for 120 min under different applied pressure of 0.1 MPa, 0.2 MPa, 0.3 MPa, 0.4 MPa and 0.5 MPa. The water flux (*J*) was measured using Equation (1), as follows
(1)$$J=\frac{V}{A\times t}\;$$
where *V* is the volume of water (L), *A* is the effective area of membrane (m^2^) and t is the filtration time. In addition, average water permeability (P_w_) was also determined to evaluate the capability of membranes to resist water flux in the applied pressure, ranging from 0.1 MPa to 0.5 MPa.

## 3. Results and Discussion

### 3.1. Morphology

Figure 1 shows the morphological structure of membranes. PD-neat membranes presented a relatively asymmetrical porous structure, as shown in Figure 1a, due to the uneven exchange rate between DMF and water during the phase inversion process. For nanocellulose-filled membranes, their porous structures had been improved after the incorporation of nanocellulose, indicating that the nanocellulose particles could induce better pore formation within the polymer matrix [21]. Furthermore, the CNW-filled membranes revealed more obvious porosity than the NCC-filled membranes. This was because the numerous reactive functional groups on the high surface area of CNWs provided it with great interfaces in PDLA polymer when compared to NCCs. In terms of nanocellulose loadings, both PDNC-I and PDNW-I membranes exhibited a more well-organized porous structure, as shown in Figure 1b,d, as compared to their corresponding PDNC-II and PDNW-II membranes, as shown in Figure 1b,e. This was probably due to the agglomeration of 2 wt.% nanocellulose particles occurring within membranes, and eventually led to the interfacial tension for macro-voids formation [22].

In addition, pore size distribution of each membrane sample is shown in Figure 2, while the average pore size values as determined from these observations is provided in Table 1. The PDNC-II membrane exhibited an average pore size close to PD-pure membrane at around 0.2 μm, revealing the poor interaction of 2 wt.% NCC filling in the PDLA matrix. However, the good porous structures were reflected by PDNC-I, PDNW-I and PDNW-II membranes, with decreased average pore sizes near to 0.13 μm. Meanwhile, the PDNW-I membrane had the smallest average pore size of 0.117 nm among these samples. Hence, based on morphological aspects, the membrane of PDNW-I demonstrates considerable higher potential for water filtration application, by attributing to its good porous feature.

In terms of surface roughness, the membranes of PD-pure and PDNW-II exhibited the great values of root mean square (R_q_) at above 0.04 (Table 1), indicating they possessed a relatively rough surface compared to other membranes. The R_q_ values had decreased for PDNC-I, PDNC-II and PDNW-I membranes at below 0.04. This implied that the membrane surface formation was smoother and more consistent for these membranes, especially for PDNW-I with the lowest R_q_ value.

### 3.2. Surface Wettability

Figure 3 illustrates captured images of water contact angles for surface wettability study, and the corresponding data was tabulated in Table 1. As seen from Figure 3a, a parabola shape of water droplet was observed to have a PD-neat membrane, implying the good surface formation of pure PDLA material. With the incorporation of nanocellulose, PDNC-I (Figure 3b), the shape of water still is able to hold a round parabola shape, similar to PD-neat, despite the hydrophilic criterion of NCC particles. The shape was likely due to the poor distribution of NCC particles on the surface of PDNC-I, and thus reduces its tendency towards water absorption [23].

However, the increment of wettability had been shown by PDNC-II (Figure 3c) with a flattened parabola shape, as well as a reduced contact angle, proving its higher affinity in adsorbing water. Besides this, both PDNW-I (Figure 3d) and PDNW-II (Figure 3e) membranes showed flattened droplet shapes and had much decreased contact angles compared to those NCC-filled membranes, by attributing this to the greatly improved hydrophilicity through CNWs incorporation [24]. Hence, it can be concluded that the CNW particles were more effective in enhancing the hydrophilicity of membranes.

### 3.3. Thermal Property

The thermal stability of membranes was evaluated by TGA analysis. From Figure 4, the TGA curve of each sample revealed an initial weight loss, in a temperature range of 60–170 °C. It was considered to be related to the vaporization of water content absorbed by the PDLA membrane. Beyond 250 °C temperature, the samples filled with nanocellulose revealed higher T_onset_ than the PD-neat sample. This indicated that the introduction of nanocellulose tends to improve the thermal heat resistance of PDLA polymer through hydroxyl groups reactions on both components. Meanwhile, the T_onset_ were lower for 2 wt.% nano-filled PDNC-II and PDNW-II samples, as compared to their corresponding PDNC-I and PDNW-I samples with 1 wt.% filler loadings (Table 2). This was likely due to a larger amount of nanocellulose with the presence of sulfate groups on the surface, which could induce an earlier thermal decomposition of PDLA molecules [12]. Furthermore, from the DTG curve, the PDNW-I showed the most stable thermal degradation among those samples, with its T_peak_ occurring at 367.4 °C. This also demonstrates that the PDNW-I membrane had promising thermal stability, which would be suitable for a water filtration application process under extreme temperatures.

### 3.4. Tensile Property

For mechanical tests, as shown in Figure 5, both T_s_ and E_ab_ was gradually increased for PDNC-I and PDNW-I samples, as compared to the PD-neat sample. The PDNC-I sample showed higher E_ab_, but with lower T_s_ than the PDNW-I sample. This was likely attributed to the effect of different pores’ connectivity between PDNC-I and PDNW-I samples, providing them with varying stress transfer properties. Additionally, the further increment of nanocellulose to 2 wt.% tremendously decreased Ts and E_ab_ for the PDNC-II sample. It was probably the existence of a large void fraction within the sample that resulted in poor stress transfer between the nanocellulose and PDLA components. Sample PDNW-II also exhibited a decrement on the values of T_s_ and E_ab_, owing to the poor porous integrity compared to the PDNW-I sample [11]. Besides this, the Y_m_ determined for those filled with nanocellulose was found to be consistent to the improvement of T_s_ (Table 2).

### 3.5. Water Permeability

Water flux through membranes applied with different pressures under continuous operation is displayed in Figure 6. In the low applied pressure of 0.1 MPa, only the PDNC-I membrane showed reduced water flux than the PD-neat. With increased pressure to 0.2 MPa, the water flux through PDNC-I exceeded the PD-neat membrane. However, from 0.2 MPa to 0.5 MPa, a fluctuated trend of up-and-down water flux occurred, which can possibly be attributed to the inconsistent pore distribution in the PDNC-I membrane. For the PDNC-II membrane, its water flux penetration dramatically increased from 0.1 MPa to 0.3 MPa and was stabilized thereafter until 0.5 MPa. The gradual increment of water flux in the 0.1–0.3 pressure range was likely promoted by the existence of large voids featured in the PDNC-II membrane. For PDNW-I and PDNW-II membranes, they presented stable water flux increment from 0.1 MPa to 0.5 MPa. From a practicability evaluation standpoint, those CNW-filled membranes possessed higher average water permeability (P_w_) of 41.36–41.92 L/m^2^h, compared to NCC-filled membranes with 39.74–40.98 L/m^2^h. This demonstrates that the membranes incorporated with CNW particles were more robust to withstand great water flux when applied under different pressures. Additionally, the CNW-filled membranes with high P_w_ prepared in this work are comparable to the multilayer polyelectrolyte nanofiltration membrane (~20–47 L/m^2^h) produced by Ouyang et al. [25], and also greater than the thermally-crosslinked nanofiltration membrane (<10 L/m^2^h) generated by Park et al. [26], in response to the separation of salts from ionic solution. Therefore, PDNW-I and PDNW-II could be regarded as reliable nanofiltration membranes for salts or metal ion removal purposes in future water treatment applications [22].

## 4. Conclusions

In this investigation, we prepared and characterized novel nanocellulose-filled PDLA membranes, verified by systematic investigations that the membranes were reliable for water filtration application. From morphology examination, membranes with a highly porous feature were formed after the incorporation of NCCs and CNWs. Besides this, the surface wettability was significantly improved for CNW-filled membranes due to the great surface reactivity of CNW particles. In addition, the thermal stability was observed to be significantly enhanced for nanocellulose-filled membranes, although the 2 wt.% loading provided a lower heat resistance than the 1 wt.% loading, which was driven by the presence of a high amount of sulfated nanocellulose. Furthermore, only the PDNW-I membrane showed improvement of mechanical property in terms of tensile strength, elongation at break and in Young’s modulus, suggesting its suitability to withstand high pressure during water filtration. This had also been demonstrated through the filtration performance evaluation, in which the PDNW-I membrane presented the highest average water permeability of 41.92 L/m^2^h under the applied pressure range of 0.1–0.5 MPa. Thus, the fabricated PDNW-I membranes could be regarded as a successful membrane product to be used for water filtration application in the future.

## Figures and Tables

**Figure 1 nanomaterials-11-00255-f001:**
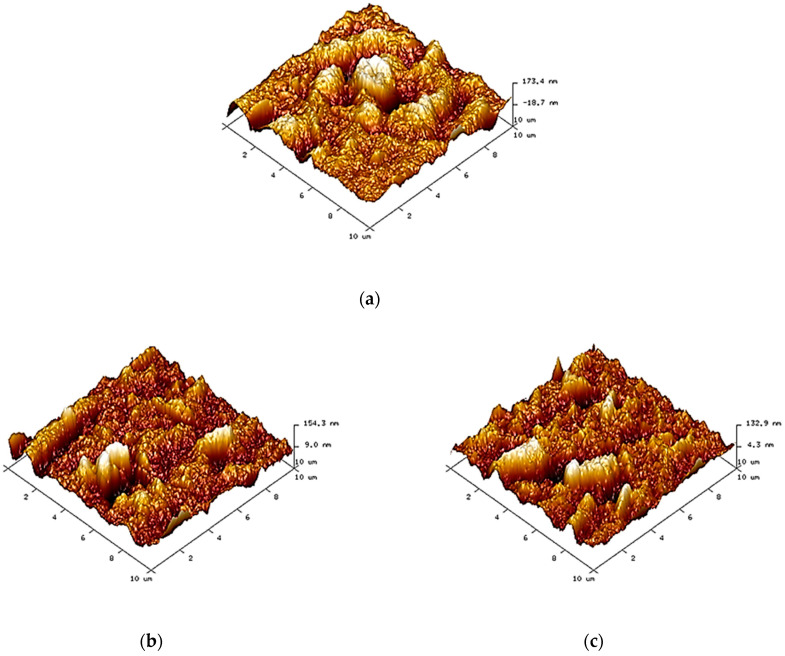
Atomic force microscope (AFM) images of: (**a**) PD-neat; (**b**) PDNC-I; (**c**) PDNC-II; (**d**) PDNW-I; and (**e**) PDNW-II membranes.

**Figure 2 nanomaterials-11-00255-f002:**
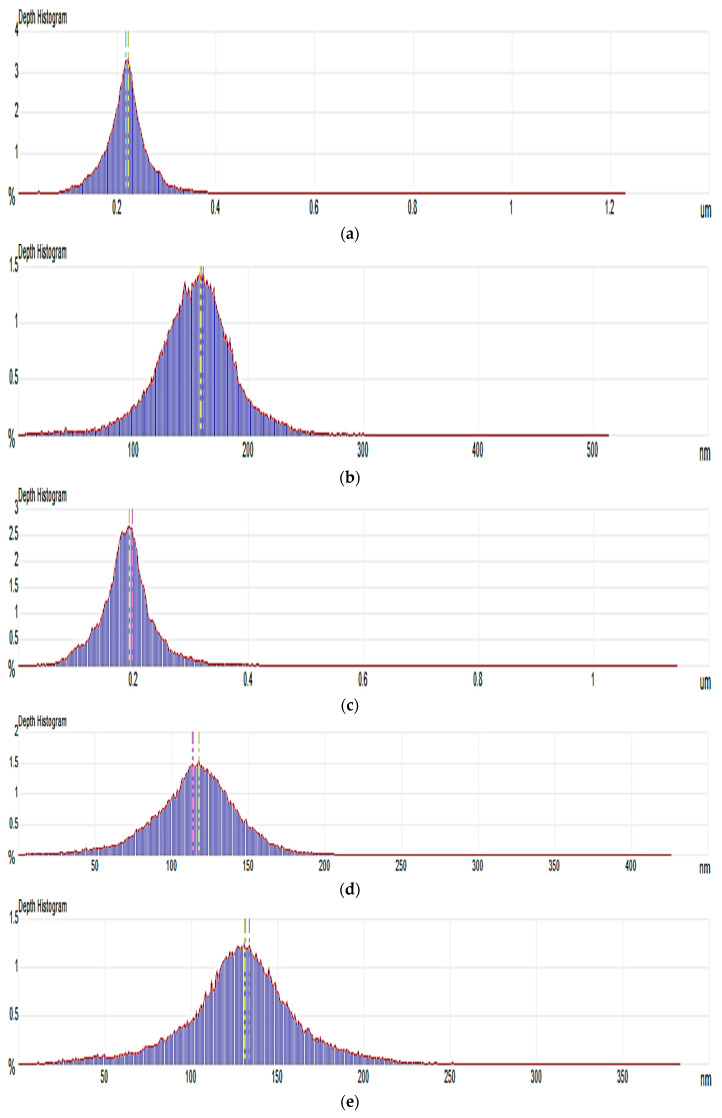
Pore size distributions of: (**a**) PD-neat; (**b**) PDNC-I; (**c**) PDNC-II; (**d**) PDNW-I; and (**e**) PDNW-II membranes.

**Figure 3 nanomaterials-11-00255-f003:**
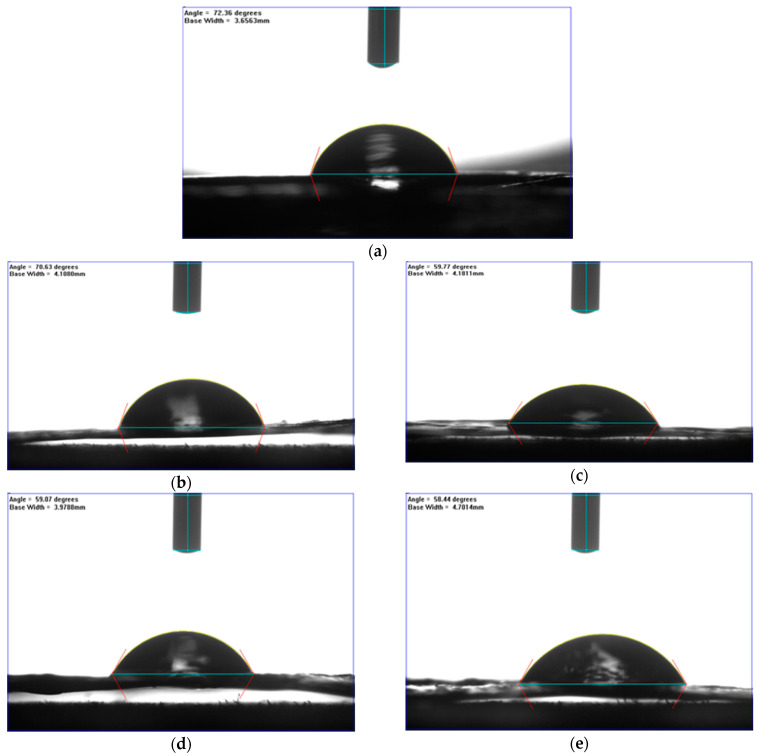
Water contact angles of: (**a**) PD-neat; (**b**) PDNC-I; (**c**) PDNC-II; (**d**) PDNW-I; and (**e**) PDNW-II membranes.

**Figure 4 nanomaterials-11-00255-f004:**
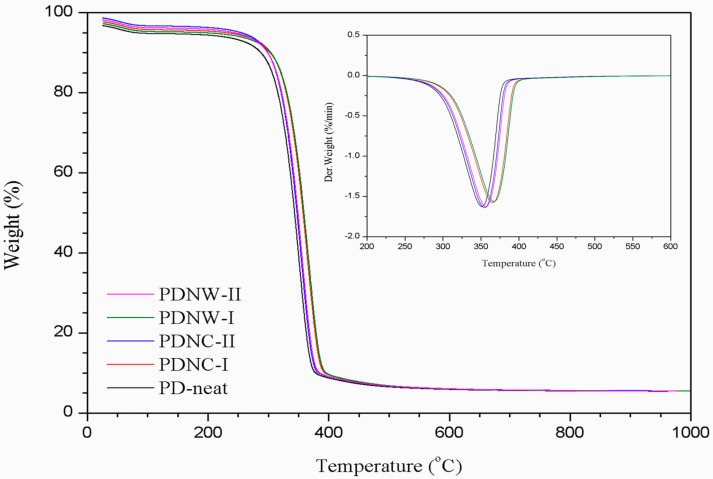
Thermogravimetric analyzer (TGA) and DTG curves of PD-neat, PDNC-I, PDNC-II, PDNW-I and PDNW-II membranes.

**Figure 5 nanomaterials-11-00255-f005:**
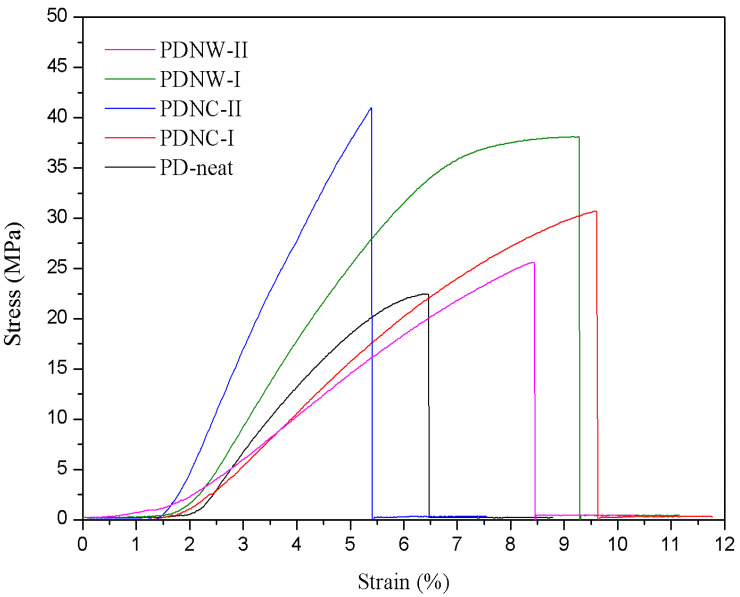
Mechanical stress-strain curves of PD-neat, PDNC-I, PDNC-II, PDNW-I and PDNW-II membranes.

**Figure 6 nanomaterials-11-00255-f006:**
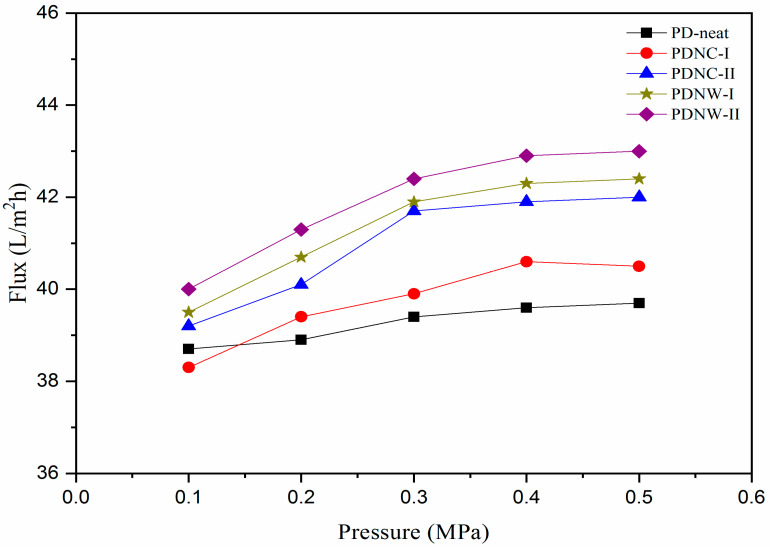
Water flux through membranes applied with different pressures.

**Table 1 nanomaterials-11-00255-t001:** Surface roughness, porosity, contact angle and water permeability data of membrane samples.

Membranes	R_q_ (nm) ^a^	S_pore_ (μm) ^b^	C_A_ (°) ^c^	P_w_ (L/m^2^h) ^d^
PD-neat	0.0479	0.220	72.36 ± 0.26	39.26
PDNC-I	0.0350	0.156	70.63 ± 0.31	39.74
PDNC-II	0.0324	0.193	59.77 ± 0.54	40.98
PDNW-I	0.0278	0.117	59.07 ± 0.32	41.36
PDNW-II	0.0435	0.134	58.44 ± 0.44	41.92

^a^ root mean square roughness; ^b^ average pore size; ^c^ water contact angle; ^d^ water permeability.

**Table 2 nanomaterials-11-00255-t002:** Thermal and mechanical data of membrane samples.

Membranes	T_onset_ (°C) ^a^	T_peak_ (°C) ^b^	T_s_ (MPa) ^c^	E_ab_ (%) ^d^	Y_m_ (GPa) ^e^
PD-neat	312.9	350.3	22.3	6.5	0.7
PDNC-I	335.9	365.9	30.5	9.6	0.5
PDNC-II	317.4	355.4	39.2	5.4	1.3
PDNW-I	336.5	367.4	37.9	9.3	0.8
PDNW-II	318.1	357.2	25.4	8.4	0.4

^a^ onset decomposition temperature; ^b^ maximum weight loss peak temperature; ^c^ tensile strength; ^d^ elongation at break; ^e^ Young’s modulus.

## Data Availability

Not applicable.

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
