# Peer review of "Fabrication and Characterization of Novel Poly(d-Lactic Acid) Nanocomposite Membrane for Water Filtration Purpose"

_nanomaterials, 2021, doi:10.3390/nano11020255_

Round 1

Reviewer 1 Report

The attempt of the authors to characterize novel membranes based on sustainable materials is worthwhile. However, the characterization presented is yet insufficient. First of all the different filtration performance and mechanical resistance  of the membranes is attributed to the generation of porosity and voids by the introduction of the cellulose fillers, but the porosity of the membranes was not measured or observed. The authors provide the BET surface areas of the fillers, what is the influence of them into the polymer matrix and as a function of the loading? 

The water absorption is difficult to discern from the drop images in Figure 2. No apparent differences are discerned among the five pictures. What is the calculated WCA from these experiments? How many pictures were taken to assure reproducibility? The water sorption or hydrophilicity could be related to the first drop in the TGA curve in Figure 3,  below 100ºC. Please complete and clarify.

The introduction could also be improved by specifically explaining the reasons why these fillers and polymer were selected in the light of the state-of-the-art literature and membrane technology advancements in membrane materials. The compatibility of stiff fillers such as NCC (line58) and crystalline polymer as PDLA (line 44) may be a challenge that the authors should envisaged to overcome. HOw? 

In section 2.1, the authors should provide the supplier and affiliation of the reactants and polymers used in their work.

line 92, Is the PDLA dried before dissolution? Also, the selection of DMF as solvent does not diminish the sustainability of the membrane fabrication process? Please justify the selection of this toxic solvent. 

Equation (1) is not the water permeability but the water flux. Please correct. 

The feed volume of 100 mL that is continuously flowed through the filtration cell seems very small for water filtration. How do the authors assure steady state in such conditions? What is the effective area of the membrane in the tests?

English should also be carefully revised.

Author Response

Reviewer #1

The attempt of the authors to characterize novel membranes based on sustainable materials is worthwhile. However, the characterization presented is yet insufficient. First of all the different filtration performance and mechanical resistance of the membranes is attributed to the generation of porosity and voids by the introduction of the cellulose fillers, but the porosity of the membranes was not measured or observed. The authors provide the BET surface areas of the fillers, what is the influence of them into the polymer matrix and as a function of the loading? 

The pore size distribution of the membranes had been analysed using AFM/Nanoscope software and added as Figure 2 inside the manuscript. Meanwhile, the average pore size was determined and added into Table 2. Besides this, we provided the BET surface area of the fillers is for reference or comparison purpose. For example, from morphology evaluation, it is valid for us to state that “the CNW filled membranes revealed more obvious porosity than the NCC filled membranes. It was because the numerous reactive functional groups on the high surface area of CNWs provided it with great interfaces in PDLA polymer when compared to NCCs.”, because we knew that the surface area of CNWs is larger than the NCCs after conducting BET analysis.

The water absorption is difficult to discern from the drop images in Figure 2. No apparent differences are discerned among the five pictures. What is the calculated WCA from these experiments? How many pictures were taken to assure reproducibility? The water sorption or hydrophilicity could be related to the first drop in the TGA curve in Figure 3,  below 100ºC. Please complete and clarify.

At the left upper corner part of drop images had indicated the water contact angles when observed in closed inspection. The analysis of water contact angles had been conducted in triplicates for each sample and this detail had been updated in the experimental section. The data of contact angles had also been added to the Table 1. Meanwhile, the statement about the vaporization of water content absorbed by PDLA membrane below 100ºC, had been added to the TGA analysis section.

The introduction could also be improved by specifically explaining the reasons why these fillers and polymer were selected in the light of the state-of-the-art literature and membrane technology advancements in membrane materials. The compatibility of stiff fillers such as NCC (line58) and crystalline polymer as PDLA (line 44) may be a challenge that the authors should envisaged to overcome. How? 

A statement of “The compatibility of nanocellulose with polymer matrix mainly depends on the Van der Waals interaction between them through hydrogen bonding.”, had been added in the introduction section in order to deliver message for overcoming the future challenge about the compatibility between nanocellulose and polymer that would be faced by other researchers.

In section 2.1, the authors should provide the supplier and affiliation of the reactants and polymers used in their work.

We added information about supplier and affiliation in the section 2.1.

line 92, Is the PDLA dried before dissolution? Also, the selection of DMF as solvent does not diminish the sustainability of the membrane fabrication process? Please justify the selection of this toxic solvent. 

The PDLA is used as purchased from supplier without undergoing any further processing to maintain the same quality conditions. This statement had been added to the section 2.1. Besides this, we used DMF because it has strong dissolution for the highly crystalline PDLA polymer and suitably applied for making good membrane structure from them. Also, since this work is our initial study for making membrane from different types of nanocellulose, we want to understand more about the ability of polymer dissolved in solvent. Nonetheless, in the future, we would modify the process by using green solvent like butanol or acetonitrile. The justification had been added to the section 2.1.

Equation (1) is not the water permeability but the water flux. Please correct. 

The phrase of ‘water permeability’ had been changed to ‘water flux’ as indicated for Equation (1).

The feed volume of 100 mL that is continuously flowed through the filtration cell seems very small for water filtration. How do the authors assure steady state in such conditions? What is the effective area of the membrane in the tests?

The feed volume was 1000 mL to continuously flow for 120 min. The effective area of membrane was 16 cm2 as stated in the section 2.5.

English should also be carefully revised.

The English had been improved throughout the manuscript.

Reviewer 2 Report

Manuscript ID: nanomaterials-1049127
Type of manuscript: Article
Title: Characterization of Poly(D-lactic acid) Nanocomposite Membrane for
Water Filtration Purpose
Authors: Lau Kia Kian, Mohammad Jawaid, Salman F. Alamery, Ashok Vaseashta

The authors of this paper have used several independent physical characterization techniques for analysis of poly(D-lactic acid) (PDLA) membranes incorporated with nanocrystalline cellulose (NCC) and cellulose nanowhisker (CNW) at different loadings of 1 wt.% (PDNC-I, PDNW-I) and 2 wt.% (PDNC-II PDNW-II) for water filtration purpose. The membranes are characterized by the atomic force microscope (AFM) and the thermogravimetric analysis (TGA). The water contact angle, the mechanical tensile and the water permeability are observed and studied. Also, the filtration performance was evaluated by allowing a 100 ml volume of water to continuously flow through a cell filtration (membrane with an effective area of 16 cm2) for 120 min under different applied pressure of 0.1 MPa, 0.2 MPa, 0.3 MPa, 0.4 MPa, 0.5 MPa.

There are many points to be modified and re-discussed critically. At the moment, this is mostly a characterization and therefore is unlikely to provide any new insight into the mechanism for improved performance. Unfortunately, without a broad and in depth data analysis, and discussion of results I can`t recommend this paper at this time. The relevant criteria set by Nanomaterials are not fulfilled. Reject.

General comments:

#

Fig. 1 Surface topography and potential mapping obtained using AFM

& Fig.3 Thermal property…

At the moment the experiments do not provide adequate information or support the rest of the findings regarding the filtration performance of the membranes and especially for PDNW-I and PDW-II (as they both showed results as reliable membranes for water filtration application).

AFM in tapping mode is a very powerful technique – on the side of the height image, the phase image should be analysed as well. The average Root Mean Square roughness of membranes, by AFM should be observed and discussed in the paper.

It is not really clear why TGA measurements were performed. It seems that the Authors have performed experiments and reported results, but not really analysed or interpreted the data.

It would be better for readers to have the filtration performance comparison with a commercially available membrane.

The level of written English should be improved a lot of Typos in the text.

Author Response

Reviewer#2

The authors of this paper have used several independent physical characterization techniques for analysis of poly(D-lactic acid) (PDLA) membranes incorporated with nanocrystalline cellulose (NCC) and cellulose nanowhisker (CNW) at different loadings of 1 wt.% (PDNC-I, PDNW-I) and 2 wt.% (PDNC-II PDNW-II) for water filtration purpose. The membranes are characterized by the atomic force microscope (AFM) and the thermogravimetric analysis (TGA). The water contact angle, the mechanical tensile and the water permeability are observed and studied. Also, the filtration performance was evaluated by allowing a 100 ml volume of water to continuously flow through a cell filtration (membrane with an effective area of 16 cm2) for 120 min under different applied pressure of 0.1 MPa, 0.2 MPa, 0.3 MPa, 0.4 MPa, 0.5 MPa.

There are many points to be modified and re-discussed critically. At the moment, this is mostly a characterization and therefore is unlikely to provide any new insight into the mechanism for improved performance. Unfortunately, without a broad and in depth data analysis, and discussion of results I can`t recommend this paper at this time. The relevant criteria set by Nanomaterials are not fulfilled. Reject.

More data related to porosity had been added into the manuscript as presented in Figure 2 and Table 1, to provide broader and deeper data analysis about membrane porosity in the results and discussion section.

General comments:

#

Fig. 1 Surface topography and potential mapping obtained using AFM …

& Fig.3 Thermal property…

At the moment the experiments do not provide adequate information or support the rest of the findings regarding the filtration performance of the membranes and especially for PDNW-I and PDNW-II (as they both showed results as reliable membranes for water filtration application).

More data related to porosity had been added into the manuscript as presented in Figure 2 and Table 1, to provide adequate information to support the filtration performance of membranes.

AFM in tapping mode is a very powerful technique – on the side of the height image, the phase image should be analysed as well. The average Root Mean Square roughness of membranes, by AFM should be observed and discussed in the paper.

It is not appropriate to add height image (2D AFM images) since it would make the manuscript looks messy because we already provided 3D AFM images. However, other data related to the average Root Mean Square roughness and pore size distribution of membranes through AFM/Nanoscope software analysis had been added, as presented in Figure 2 and summarized in Table 1, in order to improve the data variation of the manuscript.

It is not really clear why TGA measurements were performed. It seems that the Authors have performed experiments and reported results, but not really analysed or interpreted the data.

TGA analysis within manuscript improved. The TGA analysis was performed to evaluate the thermal stability of membranes. Meanwhile, some statements had been added to analyse and interpret the data.

It would be better for readers to have the filtration performance comparison with a commercially available membrane.

Since different membrane had different structures for final targeted filtration application, it is not appropriate to compare our work with commercial membrane. However, the section of water permeability analysis had been improved by comparing our membrane’s filtration performance with other reported works based on the standard of nanofiltration membrane. The comparison was also made in response to the separation of salts and metal ions from wastewater treatment, which would be conducted in our future experimental work. 

The level of written English should be improved a lot of Typos in the text.

The English had been improved and while those typo errors had been corrected throughout the manuscript.

Reviewer 3 Report

In this work, authors reported that characterization of poly(D-lactic acid) nanocomposite membrane for water filtration purpose. The authors mentioned that fabricated CNWs filled PDLA membranes have high potential to be utilized for water filtration purpose in the future. The detailed comments are given below.

  1. Figure 1: 2D AFM images of membranes are also necessary for better comparison. Scale bars in the 3D AFM images are not clearly visible.
  2. SEM images of nanocomposite membranes are required.
  3. Authors should make comparison table to compare present results with other reported literatures.
  4. Some of the following membrane related references should be cited in surface wettability section: 1016/j.compositesb.2020.107890, 10.1016/j.compositesb.2018.08.016

Author Response

Reviewer#3

In this work, authors reported that characterization of poly(D-lactic acid) nanocomposite membrane for water filtration purpose. The authors mentioned that fabricated CNWs filled PDLA membranes have high potential to be utilized for water filtration purpose in the future. The detailed comments are given below.

  1. Figure 1: 2D AFM images of membranes are also necessary for better comparison. Scale bars in the 3D AFM images are not clearly visible.

It is not appropriate to add 2D AFM images since it would make the manuscript looks messy because we already provided 3D AFM images. However, other data related to the average Root Mean Square roughness and pore size distribution of membranes through AFM/Nanoscope software analysis had been added, as presented in Figure 2 and summarized in Table 1, in order to improve the data variation of the manuscript. Also, the clarity of scale bars in 3D AFM images had been improved by inserting higher resolution images.

2. SEM images of nanocomposite membranes are required.

We are unable to provide SEM images by running additional analysis or testing in this lockdown period due to Covid-19.

3. Authors should make comparison table to compare present results with other reported literatures.

The section of water permeability analysis had been improved by comparing our membrane’s filtration performance with other reported works based on the standard of nanofiltration membrane. The comparison was also made in response to the separation of salts and metal ions from wastewater treatment, which would be conducted in our future experimental work. 

4. Some of the following membrane related references should be cited in surface wettability section: 1016/j.compositesb.2020.107890, 10.1016/j.compositesb.2018.08.016

The suggested references had been added and cited in the surface wettability section.

Round 2

Reviewer 1 Report

The manuscript has been revised and some of the doubts highlighted by the previous reviewers clarified accordingly. 

There is still concern regarding the sustainability of the membranes if the authors persist in using DMF solvent when there are others more environmentally-friendly available. The nomination of sustainable membrane materials should be removed in that case. Since apparently the authors do not have acess to SEM to improve the insight on the morphology of the membranes, deepest discussion on the results they do have (such as AFM and BET) is highly recommended. 

Regarding the behavior of the membranes, comparison with commercial membranes and filtration setups should provide the necessary stand of the new membranes in the state-of-the-art membrane technology for water filtration. 

I miss some explanation at the beginning as to the reasons for selecting these particular fillers and polymer. 

English has been slighlty improved but technical language expression should be revised by a professional native translator.

Author Response

We made changes in Manuscript as per your Suggestion. Thanks for Comments

Reviewer 2 Report

Thank you for improving the manuscript. It could be published after editing English language and style.

Next time more discussion and scientific links between different analysis methods results would be beneficial.

Author Response

Thanks for your valuable Comments. We made changes as per your suggestion
